# Alcohol Use, Anxiety and Depression among French *Grandes Écoles* Engineering Students during the COVID-19 Pandemic

**DOI:** 10.3390/ijerph20085590

**Published:** 2023-04-20

**Authors:** Marion Pitel, Olivier Phan, Céline Bonnaire, Tristan Hamonniere

**Affiliations:** 1Laboratoire de Psychopathologie et Processus de Santé, Université Paris Cité, F-92100 Boulogne Billancourt, France; marion.pitel@etu.u-paris.fr (M.P.); celine.bonnaire@u-paris.fr (C.B.); 2Centre de Soins d’Accompagnement et de Prévention en Addictologie Pierre Nicole, Croix-Rouge Française, 75005 Paris, France; 3Service d’addictologie à l’adolescence, Clinique Dupré, Fondation Santé des Etudiants de France, Sceaux, 75005 Paris, France; 4CESP Unité INSERM, 75005 Paris, France; 5Laboratoire Parallélisme, UVSQ Paris, Saclay, 75005 Paris, France; 6UR Clipsyd, Université Paris Nanterre, 92001 Nanterre, France

**Keywords:** students, alcohol, COVID-19, drinking motives, anxiety, depression

## Abstract

In French *Grandes Écoles*, heavy alcohol consumption seems to be generalized and largely tolerated, leading to particular concerns about Alcohol Use Disorder and harmful alcohol practices among students. The COVID-19 pandemic led to increased psychological difficulties, and two coexisting scenarios seemed to emerge regarding alcohol consumption: A decrease in alcohol consumption linked to the absence of festive events, and an increase in solitary alcohol consumption to cope with lockdowns. The aim of this exploratory study is to investigate the evolution of alcohol consumption, consumption motives and the relationship of these factors to the anxiety and the depression of French *Grandes Écoles* students during the COVID-19 pandemic, depending on their residential status. After the last lockdown, 353 students completed a questionnaire measuring alcohol consumption, motivation to drink, anxiety and depression during and after the COVID-19 period. Although students confined to campus were more likely to increase their alcohol use, they also presented higher well-being scores than those who lived off campus. A significant proportion of students were aware of their increased alcohol use due to the COVID-19 pandemic, and the motives attributed to their increased consumption highlight the need for vigilance and specialized support facilities.

## 1. Introduction

In France, alcohol remains the most consumed substance among youth, with many consuming to excess. Indeed, 66.5% of 17-year-olds reported drinking alcohol in the past month, with 44% disclosing an episode of binge drinking [1]. This trend continues among young adults, with more than half of 18- to 24-year-olds reporting at least one episode of binge drinking in the last year (54.1%), and the consumption of an average of 3.3 alcoholic drinks per day [2].

Among young people, students are especially vulnerable to heavy alcohol consumption [3,4,5,6]. They are also at high risk of developing Alcohol Use Disorder (AUD) [7], and harmful alcohol practices such as binge drinking, compared to their non-student peers.

This is particularly relevant among students in more demanding fields of study, such as engineering [8,9,10]. Indeed, according to the data from the Santé Publique France Health Barometer, the most important national health survey in France, in 2017, the proportion of French students reporting regular episodes of drunkenness is higher than for other young people, and in pathways such as engineering, business or management, binge drinking rates are twice as high compared to other French students [11].

If high-risk levels of alcohol consumption lead to chronic diseases over time, such as liver cirrhosis, cancer or cardiovascular disorders [12,13], the immediate consequences of such consumption can also be harmful, with the most common being missed classes, lower grades, a higher risk of injuries, overdoses, cognitive deficits, traffic accidents or sexual assaults [14,15,16,17,18,19].

Most European young adults, including students, often report social and enhancement motives for drinking [20,21,22]. In fact, alcohol consumption seems to be considered an important way to fit in for students [3,23], representing a central component of college social life [24]. However, social motives and particularly enhancement motives also seem to be linked to heavy consumption, especially among students [20,22,25]. In French *Grandes Écoles*, heavy alcohol consumption seems to be generalized and largely tolerated, leading to these students being specifically concerned by these problems [26,27,28]. Parties and massive alcohol consumption seem to constitute an important part of social inclusion in these institutions and are sometimes considered good practice for the socializing skills these students will need in their future professions [29]. 

With the outbreak of the COVID-19 pandemic, governments decided to impose lockdown measures to mitigate the spread of the virus. This general state of affairs led to an increase in psychological difficulties among young adults, including students, such as loneliness, fear, anxiety, depression and substance use [30,31,32]. Two scenarios were hypothesized regarding alcohol consumption during lockdown, and recently demonstrated to coexist in the international COVID-19 literature: A decrease in alcohol consumption linked to the absence of festive and social events [33,34,35,36,37], as well as an increase in solitary alcohol consumption, likely to cope with the impact of lockdown on anxiety and depression [33,38,39,40].

In France, three lockdowns were implemented, from 17 March to 11 May 2020; from 30 October 30 to 15 December 2020; and from 3 April 3 to 3 May 2021. The main lockdown measures that impacted students’ lives were the obligation to stay at home, the banning of non-essential travel, universities closing and online courses. The lockdowns forced a large number of students to return to their parents’ homes, while another large group had to be confined to campus. This raises the question of the impact these measures had on alcohol consumption among higher education students; as the context of these students’ lives changed over the months of lockdown, their alcohol consumption and drinking motives may have evolved.

The aim of this exploratory study is to investigate the evolution of alcohol consumption and drinking motives, and their relationship to the anxiety and depression of French *Grandes Écoles* students during the COVID-19 pandemic based on the students’ status living on or off campus.

## 2. Materials and Methods

### 2.1. Participants and Procedure

Participant recruitment took place in June 2021, one month after the last lockdown. Recruitment took place on 11 different campuses of a French engineering *Grande École*. Email invitations were sent by the school administration to all students on campus and relayed by a student association through the school’s social networks. The email invitation explained the objective and procedure of the research, as well as providing a link to the survey hosted on the LimeSurvey^®^ platform. The validity of the survey was partially controlled by preventing double participation on the LimeSurvey^®^ platform and by sending the survey link only to institutional student email addresses.

Participants who agreed to participate in the study had to provide their informed and written consent before accessing the online questionnaires. The present study was conducted in accordance with the 1964 Helsinki Declaration and its later amendments, or comparable ethical standards. A total of 378 students participated in the survey out of a total enrollment of 6000 students across the 11 campuses (6%). The response rate was 53% (378 responses for 717 views). We excluded 25 participants (6.61%) who did not consume alcohol during the 2020/2021 academic year. Therefore, a total of 353 students were included in this study.

### 2.2. Measures

#### 2.2.1. Alcohol Use

Alcohol use frequency was assessed with the first item of the Alcohol Use Disorder Identification Test (AUDIT) [41], validated in French [42]: “How often do you drink alcohol?”. Responses were keyed to a five-point Likert scale ranging from 1 (“never”) to 5 (“four or more times a week”). This question was asked for both the COVID-19 period (March 2020 to June 2021) and the period before March 2020. 

#### 2.2.2. Evolution of Perceived Alcohol Use

Eight items investigated how participants considered their alcohol use evolution (e.g., “Since March 2020, I consume higher quantities of alcohol than before”; “Since March 2020, I spend more money on alcohol than before”). Participants answered on a four-point Likert scale ranging from 1 (“totally disagree”) to 4 (“totally agree”), with a maximum possible score of 32. In our study, internal consistency was satisfactory, with a Cronbach’s alpha of 0.87.

#### 2.2.3. Drinking to Cope during the Lockdown

A group of four items was created to evaluate how each participant perceived alcohol as helping them cope during the COVID-19 period (e.g., “My alcohol use helped me worry less during the health crisis”). Participants answered on a four-point Likert scale ranging from 1 (“totally disagree”) to 4 (“totally agree”), with a maximum possible score of 16. In our study, the Cronbach’s alpha was 0.88.

#### 2.2.4. Perception of Alcohol Problems

Participants were asked to evaluate on a 10-point Likert scale how problematic they considered their alcohol consumption before March 2020 and during the COVID-19 period (from 0 “not problematic” to 10 “very problematic”).

#### 2.2.5. Drinking Motives

The motivation to drink during the COVID-19 period was evaluated using the Modified Drinking Motives Questionnaire-Revised (MDMQ-R) [43], validated in French [44]. This 28-item, self-report questionnaire is comprised of five subscales corresponding to different reasons for drinking: social motives, enhancement, conformity, anxiety coping and depression coping. Participants were asked to think about every time they drank alcohol in the past year and to rank themselves on a five-point Likert scale ranging from 1 (“never”) to 5 (“always”) for each item. Internal consistency was satisfactory, except for the anxiety subscale (α = 0.66), with a Cronbach’s alpha of 0.71 for social, 0.75 for enhancement, 0.81 for conformity and 0.94 for depression.

A question was added to evaluate if participants thought their drinking motives changed during the COVID-19 period (“Do you think your alcohol drinking motives have evolved since the emergence of the health crisis?”). If the respondent answered yes, the four main reasons to drink, along with a short explanation, were presented [45]. Then, the participant had to report how their consumption changed by answering four items, each corresponding to a reason to drink (e.g., “drink to deal with negative state/emotion”). These questions were evaluated on a five-point Likert scale ranging from 1 (“much less often”) to 5 (“much more often”).

#### 2.2.6. Anxiety and Depression

The French version [46] of the Depression Anxiety Stress Scale (DASS) [47], short form [48], was used to evaluate anxiety, depression and stress during the COVID-19 period. This 21-item scale was designed to assess the severity of the core symptoms of depression, anxiety and stress, which constitute three subscales. Each item is scored from 0 (“did not apply to me at all in the last week”) to 3 (“applied to me very much or most of the time in the past week”). A higher score indicates a higher level of distress. The form was modified to evaluate psychological distress during the past year instead of week, to better fit the needs of this study. Having modified the instruction, the cutoffs were not retained. Internal consistency was satisfactory, with a Cronbach’s alpha of 0.94 for the total score, 0.88 for stress, 0.78 for anxiety and 0.92 for depression.

The French version [49] of the Hospital Anxiety and Depression Scale (HADS) was used to investigate participant anxiety and depression in June 2021 [50]. The HADS is a 14-item, self-report scale assessing psychological distress: anxiety (seven items) and depression (seven items). All items were rated on a four-point Likert scale, with the total score ranging from 0 to 27. A score between 8 and 10 suggests the possible presence of anxiety or depression; a score of 11 or higher indicates the presence of significant symptoms of anxiety and depression. In this study, the two subscales showed good internal consistency, with a Cronbach’s alpha of 0.83 for anxiety and 0.78 for depression. 

### 2.3. Statistical Analyses

This study’s analysis was conducted on SPSS 28.0 ©. 

The normality assumption was checked through histograms, Q-Q plot, skewness and kurtosis. The normality of the AUDIT item and the social dimension of the MDMQ-R was validated, allowing us to apply t-tests, with Cohen’s d for effect size. We considered d > 0.5 as medium effect size and d > 0.8 as large effect size [51]. For the comparison of the other continuous variables, non-parametric tests using Mann–Whitney U and Wilcoxon ranks were conducted, with r value for effect size. We considered r > 0.3 as medium effect size and r > 0.5 as large effect size [52]. For categorical variables comparison, chi-square tests were conducted, with Cramer’s V for effect size, considering V > 0.3 as medium effect size and V > 0.5 as large effect size [52]. 

Thus, two groups were created, students who lived on campus during the COVID-19 period and those who lived off campus. Chi-square tests and Mann–Whitney U tests were performed to analyze socio-demographic differences between participants in the two groups. The *t*-test and the Mann–Whitney U test were used to compare mean differences of alcohol use, anxiety, depression and drinking motives between students who were locked down on or off campus. Additionally, the Wilcoxon rank test was used to compare mean differences of alcohol problem perceptions before and during the COVID-19 period. The Mann–Whitney U test was also used to compare mean differences of alcohol use, anxiety and depression between students who estimated that their alcohol consumption motivation had evolved, or not, during the COVID-19 period. 

Finally, in accordance with the objective of our exploratory study, we tested which variables were associated with psychological distress during the COVID-19 period using multiple linear regression with the backward elimination technique. In order to perform this analysis, the global score of psychological distress during the COVID-19 period (DASS total score) was chosen as the dependent variable and several variables were added to the model. To assess alcohol use, the variables alcohol use frequency, “How students estimated their alcohol use has evolved” and “How students estimated their alcohol use as problematic” were added to the model. The variable “How students estimated that alcohol helped them cope” was not retained, because of its conceptual overlap between alcohol use and motivation to drink for coping reasons. The variables residential status and drinking motives were also added. In addition to age and gender control variables, the variables student responsibility and academic year were added, following the analysis of socio-demographic differences between groups. For the gender variable, the 3 participants who answered “other” were excluded due to their low number (*n* = 3). Thus, the regression analysis was performed for 350 participants.

## 3. Results

In our sample of 353 students, 71.1% were male, with an average age of 22.05 (SD = 1.97). The sex ratio is congruent with the sex ratio of the school’s population (80% male) and the population of engineering students in France (74% male). Before the first lockdown (March 2022), 55.2% lived on campus compared to 48.2% during the COVID-19 period. Most of the students were in their first (39.7%) and second (34.6%) years of study. Due to the pandemic, 70.8% of students reported that their instruction became hybrid (in-person and online), and 14.7% responded that their instruction became totally virtual. Students living on campus were significantly younger (U = 8901; *p* < 0.001) and more likely to have student responsibility (χ^2^ = 42.0; *p* < 0.001), with medium effect size (respectively r = 0.38; V = 0.35). Sociodemographic and academic characteristics are presented in Table 1. 

### 3.1. Alcohol Use

#### 3.1.1. All Students

According to the first AUDIT item, participants consumed alcohol significantly more often during the COVID-19 period than before (t(351) = 9.08; *p* < 0.001; d = 0.48). The effect size was moderate.

#### 3.1.2. Students Living on Campus 

A comparison of alcohol use frequency among students who lived on campus showed that they consumed alcohol more often during the COVID-19 period than before (t(169) = 9.70; *p* < 0.001), with a moderate effect size (d = 0.74).

A comparison of perceived problematic alcohol use depending on period (before or during COVID-19) highlighted that students living on campus during COVID-19 indicated significantly higher perceived problematic alcohol use than before (W = 831; *p* < 0.001), with a large effect size (r = 0.50). 

#### 3.1.3. Students Living on Campus versus off Campus

Students living on campus consumed alcohol significantly more often than other students during the COVID-19 period (t(351) = 3.88; *p* < 0.001; d = 0.41), with a small effect size (see Table 2). The difference between the two groups of students before the COVID-19 period was not significant. 

The score for the perceived evolution of alcohol use highlighted that students who lived on campus during COVID-19 reported a significant increase in alcohol use compared to those living off campus, with a small effect size (U = 11993; *p* < 0.001; r = 0.20). 

Finally, students living on campus during COVID-19 presented a significantly higher score of perceived problematic alcohol use than those living off campus (U = 13446; *p* < 0.05), with a small effect size (r = 0.12). 

### 3.2. Anxiety and Depression

#### 3.2.1. All Students

After the COVID-19 period, according to the HADS scores, 19.5% (*n* = 63) had an anxiety score greater than 7. More precisely, 9.6% (*n* = 34) reported experiencing mild anxiety, 5.4% (*n* = 19) experienced moderate anxiety and 4.5% (*n* = 16) experienced severe anxiety. Likewise, 14.4% (*n* = 51) reported a depression score greater than 7. More precisely, 9.1% (*n* = 32) experienced mild depression, 4.5% (*n* = 16) experienced moderate depression and 0.8% (*n* = 3) experienced a severe presence of depression.

#### 3.2.2. Students Living on Campus versus off Campus

According to the DASS score (see Table 3), students living on campus reported significantly lower levels of distress than students living off campus during the COVID-19 period (U = 13135; *p* < 0.05), with a small effect size (r = 0.13). Students who lived on campus presented significantly lower scores for depression (U = 13032; *p* < 0.01) and stress (U = 13352; *p* < 0.05) than those who lived off campus. The effect sizes were small (r = 0.14; r = 0.12). 

After COVID-19, according to the HADS scores, students who lived on campus reported significantly lower depression (U = 13297; *p* < 0.05) and anxiety (U = 13299; *p* < 0.05) scores than those who lived off campus. The effect sizes were small (r = 0.13; r = 0.13).

### 3.3. Drinking Motives

#### 3.3.1. All Students

A small number of students (12.2%) estimated that their drinking motives had evolved due to the pandemic. Among this group, 51.2% thought that they drank more often to induce positive emotions, and 46.5% drank more often to cope with negative emotions. Moreover, 48.8% estimated that they consumed alcohol more often for social motives, and 23.3% less often. Finally, 27.9% of the students thought that they drank less often for reasons of conformity.

Students who estimated that their drinking motives had evolved during the COVID-19 period reported significantly higher depression (U = 3487; *p* < 0.001; r = 0.27), anxiety (U = 3776; *p* < 0.001; r = 0.26) and stress (U = 3256; *p* < 0.001; r = 0.29) scores, as well as a global higher level of distress (U = 2975; *p* < 0.001; r = 0.31) than their counterparts (see Table 4). The effect sizes were small, except for the distress global score, which had a moderate effect size. After the COVID-19 period, these students still reported significantly higher depression (U = 3834; *p* < 0.001) and anxiety (U = 3839; *p* < 0.001) scores than other students, with small effect sizes (r = 0.24). Even if they did not notice a change in their alcohol use frequency, these students reported an increase in the importance of their alcohol use compared to the others (U = 3383; *p* < 0.001), with a small effect size (r = 0.28). These students also indicated that alcohol helped them cope more than the other students (U = 3280; *p* < 0.001), with a moderate effect size (r = 0.32). Finally, these students estimated their alcohol use to be more problematic than the others (U = 4552; *p* < 0.001), with a small effect size (r = 0.18).

#### 3.3.2. Students Living on Campus versus off Campus

According to the MDMQ-R (see Table 5), students living on campus during the COVID-19 period drank alcohol for enhancement significantly more often than those who lived off campus (U = 13007; *p* < 0.01), with a small effect size (r = 0.14).

### 3.4. Factor Associated with Level of Distress

For the first model tested, backward elimination regression analyses indicated that gender, residential status, and coping motives (anxiety and depression) were significantly associated with distress. The four predictors contributed 40% of the variance. Age, academic year, student responsibility, other drinking motives and alcohol use variables were not significant predictors. The final model (R^2^ = 0.401) included gender (*p* = 0.003), residential status (*p* = 0.002) and coping drinking motives (*p* < 0.05) as an independent predictive factor for distress (Table 6). 

## 4. Discussion

The aim of this study was to investigate the impact of COVID-19 and living place (on or off campus) during the pandemic on alcohol use, drinking motives, anxiety and depression among a sample of French *Grandes Écoles* students. First, our results showed that all the students consumed alcohol more often during this period than before, as observed from other campus and multi-campus surveys [38,39,40]. This result contradicts other studies that demonstrated a decrease in alcohol use during the COVID-19 period [33,34,35,36,37]. These differences may be partially explained by the fact that the students’ residential situation during lockdown was not taken into consideration in some of these studies, or that campus dormitories were closed in some cases. As we did not use a standardized tool to measure alcohol consumption, we cannot compare the level of alcohol consumption between our sample and that of these studies. Nevertheless, we can see that the gender proportions are reversed: our sample is predominantly male while theirs is predominantly female [33,34,35,36,37,38,39,40].

Our results showed that students who lived on campus consumed alcohol more often than those who lived off campus during the COVID-19 period, with the difference before the lockdown not significant. Furthermore, we showed that on-campus students indicated an increase in the importance of their alcohol use, reporting higher scores of perceived problematic use than those who lived off campus. These results are consistent with the literature, which highlights that students are particularly exposed to the risk of developing AUD [7] and harmful alcohol practices [8,9], particularly among *Grandes Écoles* students [29]. Alcohol seemed to remain an important part of student life on campus, even during the pandemic [24]. This result can be paralleled with the higher number of participants with student responsibility on campus than off campus. These students, who are more involved in college life, should therefore also be more exposed to parties where heavy drinking is widespread [26,28,29]. Similarly, students on campus were younger than students off campus: these students reported drinking more often, larger amounts of alcohol, spending more money on it, having more problems with it, etc. This is consistent with the prevalence in France, which show that the younger an individual, the more heavy drinking, such as binge drinking, they do [2].

However, we also showed that students living on campus reported significantly lower levels of psychological distress than those living off campus during the COVID-19 period and, more specifically, lower scores of depression and stress than those living off campus. The regression analysis results support this observation. Residential status is a significant predictor of the level of psychological distress during the year. We also demonstrated that this effect persisted, with students living on campus reporting lower levels of depression and stress than their counterparts after the end of the COVID-19 period. One hypothesis could be that staying together offered a protective factor against psychological distress. Other studies of the impact of COVID-19 showed that loneliness led to an increase in symptoms of depression and anxiety [32]. In France, for instance, a national longitudinal survey of the several lockdowns during the COVID-19 period showed that young people felt particularly isolated, including when they returned to confine themselves in their family home, and that young men (18–25 years) were the most affected by emotional distress, especially in overcrowded housing [53,54,55]. Moreover, students on campus seem to have been spared many of the difficulties of returning to the family home (lack of an unshared workspace, deterioration of parent–child relationships) [53,56].

Therefore, we might deduce that being locked down together on campus prompted students to continue social alcohol use habits without impacting alcohol motivation. Indeed, it is interesting to note that the mean social dimension score obtained by our student population remained higher than the other drinking motive dimensions and is close to the one obtained by Grant et al. [43] from two other student samples outside of any pandemic situation. This implies that the pandemic did not impact the social motivation for alcohol consumption among students. Moreover, some students shared in the comment space at the end of the study that “The lockdowns, with online courses and everything, ultimately just increased opportunities for parties, but I don’t think it’s that decisive”; “The lockdown setting for students facilitated consumption, but without any particular problems.” Consideration of the proximity of student bedrooms, communal areas and the self-management of student residences suggests that the students had an active social life during confinement. Those results could explain the fact that our student population presented a particularly low level of psychological distress during and after the COVID-19 period. Indeed, even though we cannot use the DASS cutoff to estimate the level of psychological distress of students during the COVID-19 period, the scores remain very low. In addition, the majority of the students have low anxiety and depression scores after the COVID-19 period, which means that they still have a low level of psychological distress after the COVID-19 period.

Furthermore, there was no difference in social drinking motivation levels between the two student residential groups. Two elements could explain this lack of difference: new consumption habits, such as the virtual happy hours that appeared during the lockdowns, and a general increase in alcohol consumption that seems to have affected the population as a whole [32,57,58]. 

However, a non-negligible number of students presented a probable risk of anxiety disorder; this prevalence is lower than results obtained from similar student populations outside of any pandemic situation [59,60]. Likewise, a non-negligible number of students presented a probable risk for depression. This prevalence seems to be similar, on average, to that in other studies of a similar population outside of any pandemic situation [59,60]. These results are in line with previous ones as they can lead to the impression that our population has particularly low levels of psychological distress. However, this engineering French *Grandes Écoles* population has the particularity of being privileged [29], which seemed to constitute an important protective factor against psychological distress during the COVID-19 period [55]. This may explain how our sample seems to be protected from the psychological distress that has affected the student population so much [30,31,32,55].

The results also showed that students on campus consumed alcohol for enhancement more often than those off campus. Alcohol consumption, as a means of having fun, perhaps became a way to cope with the boredom induced by lockdowns, as demonstrated in a report published by Nanos Research in Canada, where participants reported drinking more alcohol during the COVID-19 period due to boredom [61], or in another study in Austria, where participants consuming alcohol during the COVID-19 period were more affected by boredom [62]. 

Furthermore, a significant proportion of students estimated that their alcohol consumption evolved during the pandemic. Among this group, half thought that they drank more often to induce positive emotions, and almost half of this group drank to cope with negative emotions. This result is very important, as difficulties with emotion regulation are one of the main factors involved in the initiation and development of AUD [63,64,65,66,67]. Thus, we could think that a significant part of our studied population is more at risk for developing an AUD to cope with the pandemic situation. In our study, students who estimated that their alcohol consumption evolved during and after the COVID-19 period reported an increase in their alcohol consumption, and estimated that their alcohol consumption was more problematic. They also reported higher scores of depression, anxiety and stress, as well as a globally higher level of distress than students who estimated that their alcohol consumption did not evolve. Yet, we also showed that these students estimated that alcohol helped them cope more than others. These results are in line with previous studies, in which alcohol use was found to be a way of coping with psychological difficulties induced by the pandemic [68]. It would be interesting to investigate if these results are maintained once the pandemic situation has been resolved.

This study comes with several limitations. First, the cross-sectional and retrospective nature of the study hinders any causality statement. Second, the use of different questionnaires to assess anxiety and depression during and after the COVID-19 period prevented further statistical analyses. This choice was nevertheless made to promote better completion of the questionnaires, avoiding different biases that could have been strong with repeated questions. The same is true for the decision to use only a partial AUDIT, which could have been a source of additional results. Third, because part of the assessment of alcohol use and symptoms of anxiety and depression was over a 12-month recall period, participants’ responses may have been subject to recall bias. Finally, different degrees with different levels of requirements have not been compared.

## 5. Conclusions

While students confined to campus were more likely to increase their alcohol use, they also presented greater well-being scores than those who lived off campus. Furthermore, a significant proportion of students were aware of their increased alcohol use due to the COVID-19 pandemic, and the motives attributed to their consumption highlight the need for specialized health care centers where students can talk about their consumption. Since those students presented problematic alcohol use and psychological difficulties, they were probably more at risk for developing psychological disorders. French *Grandes Écoles* must be vigilant about this issue and should implement appropriate support systems. More generally, these results should be taken into consideration in public health strategies and preventions for future pandemics or other stressful life events.

## Figures and Tables

**Table 1 ijerph-20-05590-t001:** Sociodemographic and academic characteristics of the sample (*n* = 353).

Characteristics	*n* (%) or m (SD)	On Campus during the COVID-19 Period (*n* = 170; 48.2%) *n* (%) or m (SD)	Off Campus during the COVID-19 Period (*n* = 183; 51.8%) *n* (%) or m (SD)	Chi-2 Test or Mann–Whitney U Test
Gender				χ^2^ = 0.013 *p* = 0.909
Female	99 (28)	48 (48.5)	51 (51.5)	
Male	251 (71.1)	120 (47.8)	131 (52.2)	
Other	3 (0.8)	2 (66.7)	1 (33.3)	
Age	22.1 (1.97)	21.4 (1.24)	22.7 (2.29)	U = 8901 *p* < 0.001
Student responsibility	219 (62)	135 (61.6)	84 (38.4)	χ^2^ = 42.0 *p* < 0.001
Academic year				χ^2^ = 26.35 *p* < 0.001
1		88 (62.9)	52 (37.1)	
2		56 (45.9)	66 (54.1)	
3		25 (30.5)	57 (69.5)	
4 and more		1 (11.1)	8 (88.9)	

**Table 2 ijerph-20-05590-t002:** Alcohol consumption before and during the COVID-19 period, depending on residential status (*n* = 353).

Students on Campus versus off Campus during the COVID-19 Period	**Total** **(*n* = 353)**	**On Campus** **(*n* = 170)**		**Off Campus** **(*n* = 183)**	**Mann–Whitney U Test or *t*-Test**
	**m (SD)**	**m (SD)**		**m (SD)**	
Alcohol use frequency	2.53 (0.99)	2.74 (1.01)		2.34 (0.94)	t (351) = 3.88 *p* < 0.001
How students estimated their alcohol use has evolved	13.2 (5.15)	14.2 (5.29)		12.3 (4.85)	U = 11993 *p* < 0.001
How students estimated that alcohol helped them cope	5.90 (2.86)	5.97 (2.87)		5.84 (2.86)	U = 14986 *p* = 0.503
How students estimated their alcohol use as problematic	2.85 (2.46)	3.15 (2.45)		2.58 (2.44)	U = 13446 *p* = 0.025
Focus on students on campus (*n* = 170)	**Before COVID-19 period**		**During COVID-19 period**		***t*-test or Wilcoxon rank**
	**m (SD)**		**m (SD)**		
Alcohol use frequency	1.86 (1.11)		2.74 (1.01)		t (169) = 9.70 *p* < 0.001
How students estimated their alcohol use as problematic	1.95 (2.29)		3.15 (2.45)		W = 831 *p* < 0.001

**Table 3 ijerph-20-05590-t003:** Comparison of student anxiety and depression depending on residential status and period (*n* = 353).

	Total (*n* = 353)	On Campus (*n* = 170)	Off Campus (*n* = 183)	Mann–Whitney U Test
	m (SD)	m (SD)	m (SD)	
DASS (during lockdown)	11.03 (11.55)	9.34 (10.48)	12.61 (12.28)	U = 13135 *p* = 0.011
Depression	4.81 (5.23)	3.99 (4.59)	5.58 (5.67)	U = 13032 *p* = 0.008
Anxiety	2.14 (3.15)	1.82 (2.80)	2.45 (3.43)	U = 14632 *p* = 0.315
Stress	4.08 (4.45)	3.54 (4.21)	4.58 (4.63)	U = 13352 *p* = 0.020
HADS (2021 June)				
Depression	3.67 (3.38)	3.22 (3.13)	4.09 (3.56)	U = 13297 *p* = 0.017
Anxiety	5.09 (4.00)	4.62 (3.84)	5.52 (4.11)	U = 13299 *p* = 0.018

**Table 4 ijerph-20-05590-t004:** Comparison of student anxiety, depression and alcohol consumption between periods depending on perceived evolution of alcohol use motivation (*n* = 353).

	Alcohol Consumption Motivations Evolved (*n* = 43)	Alcohol Consumption Motivations Did Not Evolve (*n* = 310)	Mann–Whitney U Test
	Mean (SD)	Mean (SD)	
DASS (during lockdown)	20.98 (12.70)	9.65 (10.70)	U = 2975 *p* < 0.001
Depression	8.91 (5.89)	4.25 (4.88)	U = 3487 *p* < 0.001
Anxiety	4.58 (4.26)	1.81 (2.82)	U = 3776 *p* < 0.001
Stress	7.49 (4.68)	3.60 (4.22)	U = 3256 *p* < 0.001
HADS (2021 June)			
Depression	6.35 (4.37)	3.30 (3.05)	U = 3834 *p* < 0.001
Anxiety	8.19 (5.13)	4.66 (3.63)	U = 3839 *p* < 0.001
Alcohol use			
Alcohol use frequency during the COVID-19 period	2.79 (0.80)	2.50 (1.01)	U = 5693 *p* = 0.103
Alcohol use evolution	17.74 (6.18)	12.60 (4.67)	U = 3383 *p* < 0.001
How students estimated alcohol helped them cope	8.51 (3.60)	5.54 (2.55)	U = 3280 *p* < 0.001
How students estimated their alcohol use as problematic during the COVID-19 period	4.14 (2.61)	2.67 (2.38)	U = 4552 *p* < 0.001

**Table 5 ijerph-20-05590-t005:** Drinking motives depending on residential status during the COVID 19 period (*n* = 353).

Drinking Motives	Total (*n* = 353)	On Campus (*n* = 170)	Off Campus (*n* = 183)	Mann–Whitney U Test or *t*-Test
	m (SD)	m (SD)	Mean (SD)	
Social	2.79 (0.85)	2.84 (0.84)	2.73 (0.85)	t(351) = 1.235 *p* = 0.218
Enhancement	2.21 (0.95)	2.34 (0.96)	2.08 (0.93)	U = 13007 *p* = 0.007
Conformity	1.23 (0.47)	1.23 (0.43)	1.24 (0.50)	U = 15020 *p* = 0.515
Anxiety coping	1.65 (0.69)	1.68 (0.72)	1.61 (0.67)	U = 14742 *p* = 0.389
Depression coping	1.34 (0.70)	1.30 (0.65)	1.37 (0.74)	U = 15206 *p* = 0.683
**Drinking motives evolution** Among students who estimated that their alcohol consumption motivations evolved during the COVID-19 period (*n* = 43)	**Induce positive emotional states**	**Face negative emotions**	**Social motives**	**Conformity motives**
Less often	*n* = 7 16.3%	*n* = 8 18.6%	*n* = 10 23.3%	*n* = 12 27.9%
Not less often or more often	*n* = 14 32.6%	*n* = 15 34.9%	*n* = 12 27.9%	*n* = 25 58.1%
More often	*n* = 22 51.2%	*n* = 20 46.5%	*n* = 21 48.8%	*n* = 6 14.0%

**Table 6 ijerph-20-05590-t006:** Factor associated with level of psychological distress (multiple linear regressions, backward elimination technique) (*n* = 350).

	Model 1	Model 9
Gender	β = −0.121 *p* = 0.007	β = −0.126 *p* = 0.003
Age	β = 0.046 *p* = 0.347	
Residential status	β = 0.103 *p* = 0.035	β = 0.128 *p* = 0.002
Academic year	β = 0.017 *p* = 0.717	
Student responsibility	β = −0.005 *p* = 0.923	
Drinking motives		
Social	β = 0.004 *p* = 0.945	
Enhancement	β = −0.013 *p* = 0.829	
Conformity	β = 0.083 *p* = 0.088	β = 0.083 *p* = 0.059
Anxiety coping	β = 0.142 *p* = 0.039	β = 0.136 *p* = 0.026
Depression coping	β = 0.460 *p* < 0.001	β = 0.458 *p* < 0.001
Alcohol use		
Alcohol use frequency	β = −0.042 *p* = 0.455	
How students estimated their alcohol use has evolved	β = 0.067 *p* = 0.247	
How students estimated their alcohol use as problematic	β = −0.054 *p* = 0.374	
Variance explained by model	R^2^ = 0.408	R^2^ = 0.401
Statistical significance of model	F (13, 336) = 17.820 *p* < 0.001	F (5, 344) = 46.048 *p* < 0.001

## Data Availability

The data presented in this study are available on request from the corresponding author.

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
