# Peer review of "Alcohol Use, Anxiety and Depression among French Grandes Écoles Engineering Students during the COVID-19 Pandemic"

_ijerph, 2023, doi:10.3390/ijerph20085590_

Round 1

Reviewer 1 Report

 Alcohol use and mental health among French Grandes Écoles 2 engineering students during the COVID-19 pandemic 

 The introduction is very organized and gives us a good picture of the problem. The methodology is very well described. The results and discussion section needs to be improved. 

 The students were under restrictions at the time of the survey and questions ask them to remember the previous time.  how do you manage recall bias? 

Could you please provide the response rate? (response/ view) 

Why "gender" was not included in the introduction and in further analysis?

The discussion could be written more in-depth. It is more about presenting the previous studies instead of discussing them. 

Also, some issues can be addressed below, as follow:

line 39 - is this not also relevant that male students have more alcohol consumption than females and this course has in general more male than female students?  This could be addressed in the French context

line 89 - was this study approved by ethical comitté? Needs to have the institution that approved and the number.

line 151 - SPSS is a registered brand and needs to be acknowledged

line 320 - previous studies, please had a few more references

Author Response

Dear editors, dear reviewers,

We were really pleased to receive the review of our manuscript and to learn of your editorial decision to consider a revised version of this manuscript for publication in IJERPH journal.

We were also grateful to take note of the comments and recommendations that have been made by the Reviewers as they allowed us to improve the quality of our paper. Therefore, please find attached the revised version of the manuscript where changes are highlighted in yellow.

We thank you for your interest in our research and look forward to hearing from you.

Sincerely

Marion Pitel,  Olivier Phan, Céline Bonnaire, Tristan Hamonniere

Reviewer 2 Report

I found this study very interesting and timely. However, few suggestions below,

- Please provide rationale on selecting Engineering students as samples

- 'Mental health' is a broad term, instead suggesting use stress, anxiety, depression term specifically. Substance use disorder (e.g., alcohol) is also considered as an mental health problem. 

- Please describe data collection procedure

- Since the authors aimed to "investigate the evolution of alcohol consumption and motives for drinking, and their relationship to the mental health of French Grandes Écoles students during the COVID-19 pandemic based on the students’ status living on or off campus', why the authors did not ran regression models? Please explain! I would suggest to re-do the analyses to rectify the association between independent and outcome variables. 

- Also need to update the discussion and conclusion based on the new analysis. Currently, the descriptive statistics do not provide any new insights. 

Author Response

(The authors gave the same response as above.)

Reviewer 3 Report

The subject is of interest, especially from the point of view of public health (44% revealed an episode of excessive alcohol consumption) and from the social and academic point of view.

A complete and updated theoretical review of the subject is presented. The background and theoretical foundation are consistent with the objective that is set. The objectives are clear.

The exploratory study was conducted in a sample possibly vulnerable to excessive alcohol consumption and mental health problems.

In relation to the methodology, it is not explained if some type of sampling was carried out to assess the representativeness of the sample or if, on the contrary, it is a convenience sample. If this has not been done, it would be convenient to include the total population. The number of French Grandes Écoles is not known, nor is the total number of engineering students on the 11 campuses.

Also, the procedure that was carried out to obtain the data should be explained in more detail.

The evaluation instruments present adequate psychometric characteristics.

The treatment and the statistical analyzes are adapted to the objectives.

The results are presented clearly through tables. The discussion is consistent with the objectives and results.

Finally, a reasoned and justified approach to the importance of the findings is made and the implications, limitations and conclusions of this study are described. In limitations, it is suggested to include that different degrees with different levels of requirements have not been compared.

Author Response

(The authors gave the same response as above.)

Round 2

Reviewer 1 Report

Dear Authors,

I would like to express my gratitude for your prompt response to my feedback and the improvements made to your paper. Your efforts in addressing my concerns are highly appreciated and have significantly enhanced the quality of your work.

I am glad to see that you have taken my suggestions into consideration and made the necessary revisions. Your attention to detail and commitment to excellence are evident in the revised version of your paper.

Thank you again for your hard work and dedication to this project.

Author Response

Thank you for your comments.